# An Omnidirectional Near-Field Comprehensive Damage Detection Method for Composite Structures

**Zhiling Wang [1,\*], Zhenwei Xiao [2,\*], Yonglin Li [1] and Yudong Jiang [1]**

1   School of Mechanical and Electrical Engineering, Jinling Institute of Technology, Nanjing 2211169, China; eitie@jit.edu.cn (Y.L.); jasonjiang_d@163.com (Y.J.)
2   Nanjing Ericsson Panda Communication Co. Ltd., Nanjing 2211169, China
\*   Correspondence: wangzhiling2013@nuaa.edu.cn (Z.W.); david.h.xiao@ericsson.com (Z.X.)

**Abstract:** As one of the active structural health monitoring methods based on the Lamb wave, the ultrasonic phased-array damage detection method can provide information such as damage location and range more intuitively, which is why this method is a research hotspot in the field of Lamb wave-based damage monitoring. However, the ultrasonic phased-array damage detection method intended for the far field is not applicable to near-field damage monitoring. In addition, the traditional one-dimensional piezoelectric phased-array damage imaging method suffers from a blind area in the near field, and the data collection time of its angle scanning is relatively long. In view of these problems, this paper proposes an omnidirectional damage imaging monitoring method, combining the near-field sampling phased-array damage monitoring algorithm and the two-dimensional phased-array. The proposed method is verified by experiments using complex composite materials, and the results obtained show that the proposed omnidirectional near-field sampling phased-array damage imaging method is suitable for real-time damage detection in complex composite materials.

**Keywords:** structural health monitoring; ultrasonic phased array; composite structure; complex structures

---

## 1. Introduction

Compared with traditional metal materials, composite materials have higher specific strength, specific stiffness, and strong designability, which is why they are widely used in the aerospace industry. However, the process of forming composite material-based components is extremely complex and there are many factors affecting the performance, namely, small differences in process parameters that can cause many defects. The composite structure serving process is vulnerable to many factors that may cause internal damage, such as hail impact, birds, and lightning effects. Sometimes damage cannot be detected in the early stages of production or in a timely manner and this can cause damage accumulation, which can further result in a significant decrease in structural strength and stability, and the safety of the aircraft structure could be seriously affected and its service life also be significantly shortened [1–3]. Therefore, it is of great significance to monitor structural health and identify damage in an aviation composite structure in a timely manner, and this requires the application of a structural health monitoring method.

Ultrasonic guided waves have the advantage of long propagation distance, energy concentration, and convenient excitation/collection in plate and shell structures. With the extensive application of composite structures in the aerospace industry, higher attention has been paid to the application of ultrasonic guided wave-based health monitoring technology in the wing skin, connection structure, fuel tank, and similar structures. The ultrasonic phased-array damage imaging method is an active

structural health monitoring method based on a Lamb wave. By controlling the phase delay of a damage scattering signal received by a piezoelectric sensor array, the method can realize the directional scanning of a structure. Moreover, a damage scattering signal can be enhanced so that the signal-to-noise ratio of the damage signal can be improved, which is applicable to the identification of small damages in a slab structure, such as cracks and delamination.

Giurgiutiu et al. first introduced the phased-array into piezoelectric elements to monitor plate structure [4]. Wilcox [5], Malinowski [6], Yuan [7], and Yu [8] studied the basic principles of Lamb waves and ultrasonic phased-array technology and verified by experiment the practical application of this technology in aviation structural health monitoring and damage detection.

A one-dimensional linear ultrasonic phased-array transducer can only monitor well the structural area between 0° and 180°, while damage that is closer to 0° or 180° will be difficult to identify due to the existence of a blind spot monitoring angle. The mentioned limitation of linear arrays can be overcome by using two-dimensional (2D) arrays. Giurgiutiu and Yu et al. proposed a two-dimensional phased-array, which can solve a larger error problem when damage is close to the array, and the omnidirectional scanning of a slab structure can be realized [9]. Malinowski et al. proposed an improved form of a mi-zi array and combined multiple linear phased-array imaging results to realize crack monitoring of aluminum plates [6]. Yoo and Purekar studied the phased-array algorithm based on a spiral array and realized crack monitoring in aluminum plates and composites [10,11]. Yuan and Wang studied structure health monitoring based on a two-dimensional linear phased-array ultrasound and realized an effective identification of multiple damages in an aluminum structure used in aviation [12]. Papulak et al. applied phase array technology in Air Traffic Control aerospace structures to develop a set of active and passive compatible structural health monitoring systems, which were then used to determine damages in large carrier rocket composite structures [13].

However, the research on the application of the ultrasonic phased-array technique in structural health monitoring of composite materials is still not mature, and there are still many difficulties and problems that need to be studied further. For instance, the piezoelectric element beam directivity in an ultrasonic phased-array transducer can be thought of as parallel in the near-field, but there is no obvious directivity; also, the signals of the piezoelectric elements cause interference and influence each other, and it is difficult to get an accurate damage identification due to the formation of the near-field blind area. Moreover, the anisotropy and structure of the composite materials have a complicated form, which also introduces a lot of difficulties to the application of ultrasonic phased-array technology.

In this paper, a sampling of the phased-array technique for near-field damage monitoring is proposed.

The rest of the paper is organized as follows: in Section 2, the omnidirectional near-field comprehensive damage detection method is presented; in Section 3, the proposed method is validated by the experiments on a complex composite fuel tank; in Section 4, an analysis of the experimental results is provided; and finally, the conclusion is given in Section 5.

## 2. Ultrasonic Phased-Array Damage Imaging Method

### 2.1. Definition of Near Field and Far Field

In the traditional sonar theory, the far-field region is defined by [14]:

$$R_{far} > \frac{2D^2}{\lambda} \tag{1}$$

where $D$ is the length of a piezoelectric array, and $\lambda$ is the wavelength of a Lamb wave generated by the excitation signal. In the ultrasonic phased-array field, the far-field can be regarded as a region at a much larger distance from the phased-array than the array length, $D$. On the one hand, when a target is in the far-field region, the excitation signal wave of the one-dimensional array elements is approximately a parallel wave, and the propagation direction of every piezoelectric element wave

is approximately parallel to that of the received Lamb wave signal; then, beam synthesis can be conducted by using the parallel theory of an ultrasonic phased-array.

On the other hand, the near-field region is defined by:

$$0.62\sqrt{\frac{D^3}{\lambda}} < R_{near} \le \frac{2D^2}{\lambda} \tag{2}$$

In the region with the diameter of $R \le 0.62\sqrt{\frac{D^3}{\lambda}}$, the phased-array theory does not hold. Additionally, when a target is in the near-field region, as shown in Figure 1, the propagating wavefront is curved (a circular wavefront), and the wave propagation direction depends on the position of each element of a one-dimensional array.

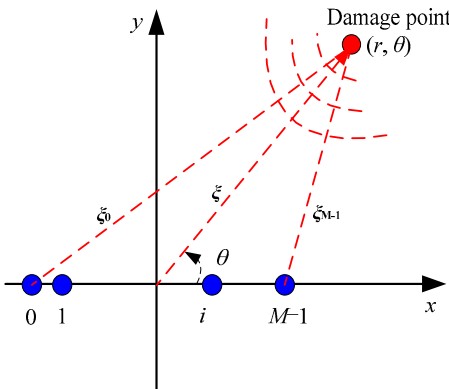

**Figure 1.** The near-field wavefront diagram of a one-dimensional array.

In the near-field region, if the damage is located by using the parallel wave algorithm intended for the far-field, the result will either contain a large error or the damage will not even be able to be identified. This is because a one-dimensional phased-array ultrasound suffers from near-field blindness.

*2.2. Near-Field Sampling Ultrasonic Phased-Array Damage Imaging Method*

The near-field sampling principle of the proposed ultrasonic phased-array imaging method is as follows: every position in the near-field region is considered as a virtual point source, and the pixel value of every point source is considered as an emission transducer array synthesis amplitude of the received signal component; the pixel values of all points are displayed in image form, and then the structural damage imaging process is conducted, where the highest-energy pixels are judged to be a damage position $P(r, \theta)$. The principle of sampling near-field phased-array ultrasound imaging is presented in Figure 2.

Then, the sampling phased-array data acquisition method is adopted, where the data collection process and sensing time determination of the transducer array elements are performed without distinction between the angle acquisition signal and the damage scattering signal, which is determined using the time delay given by Equation (3). The scanning cycle of this method is short, which can improve the efficiency of data acquisition and processing analysis, so a real-time data assessment is possible.

Where the center of a linear array is at the coordinate origin, and M piezoelectric elements represent the transducer array with the array number M; the $i$th piezoelectric element excitation signal received at the $j$th piezoelectric element is labeled as $A_{ij}(t)$, $(i, j = 0, 1, 2, \ldots, M - 1)$. When all piezoelectric elements are excited one by one, and when one element is excited, the other piezoelectric elements act as receivers, so that a complete data set $A_{ij}(t)$, $(i, j = 0, 1, 2, \ldots, M - 1)$ can be obtained.

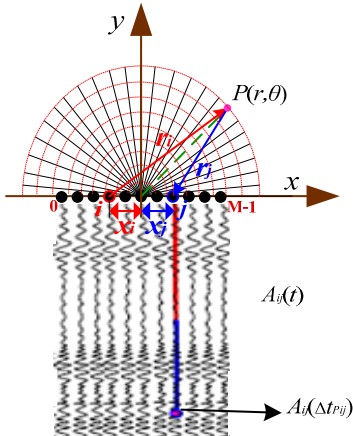

**Figure 2.** The principle of sampling near-field phased-array ultrasound imaging.

Any point source in the monitoring area can be synthesized by using the transmitting and receiving signals from various piezoelectric elements in the transducer array, which provides a theoretical basis for the generation of damage imaging monitoring. As shown in Figure 2, for the transmitting piezoelectric element *i* and the receiving piezoelectric element *j* at the respective distance from the point $P$ $r_i$ and $r_j$, the time $t_{pij}$ the signal needs to reach $P$ with the ultrasonic wave velocity $c$ can be expressed as $(r_i + r_j)$, then the focused energy component $A_{Pij}(t_{pij})$, $(i, j = 0, 1, 2, \ldots, M-1)$ is delayed according to the energy superposition principle of a phased-array, which is given by:

$$\Delta t_{Pij} = \frac{r_i + r_j - 2r}{c} = \frac{\sqrt{r^2 + x_i{}^2 - 2rx_i \cos\theta} + \sqrt{r^2 + x_j{}^2 - 2rx_j \cos\theta} - 2r}{c} \tag{3}$$

$$x_i = (i - \frac{M-1}{2})d \tag{4}$$

$$x_j = (j - \frac{M-1}{2})d \tag{5}$$

where $x_i$ and $x_j$ are the distances between the *i*th and *j*th piezoelectric elements from the coordinate origin, respectively. According to Equation (3), the time delay in the near-field can be used to determine the polar angle $\theta$ (in a one-dimension linear array $\theta \in [0°, 180°]$) and the pole size $r$, where the step of the polar angle is 1°, and the step of the polar size is 1 mm. The delayed near-field focusing energy component $A_{Pij}(\Delta t_{pij})$, $(i, j = 0, 1, 2, \ldots, M-1)$ is then superimposed, and finally, the pixel size at point $P$ in the direction of angle $\theta$ is given by:

$$S_P = \alpha \sum_{i,j=0}^{M-1} A_{ij}(\Delta t_{pij}) \tag{6}$$

where $\alpha$ is the signal conversion coefficient. In this work, $\alpha$ is taken to be 1, regardless of the influence of a distance on the waveform. By analogy, in a detection area from 0° to 180° direction, the virtual point source with the biggest amplitude in the angle θ direction given by Equation (7) represents a damage point $P$.

$$S_\theta = \max\left(\sum_{i,j=0}^{M-1} A_{ij}(\Delta t_{pij})\right) \tag{7}$$

Using the ultrasonic phased-array technology, the position of damage can be determined using the angle $\theta$ and radius $r$, which can be calculated by using the signal arrival time $\Delta t$ and Lamb wave velocity $c$; the distance $r$ of the damage from the coordinate system origin is given by:

$$r = \frac{\Delta t \cdot c}{2} \tag{8}$$

The signal energy (amplitude) can be expressed as a function of its distance and direction, which represents a function of time and angle. Meanwhile, the scanning image can be obtained by displaying the energy of a synthetic signal, so the damage can be clearly displayed in the scanning image.

To make the damage position more intuitive and easier to observe in the image, its polar coordinates, namely angle $\theta$ and radius $r$, are converted into Cartesian coordinates. Assuming that $x$ and $y$ are respectively the horizontal and vertical coordinates in the Cartesian coordinate system, the corresponding coordinates conversion relation is given by:

$$\begin{cases} x = r \cos \theta \\ y = r \sin \theta \end{cases} \tag{9}$$

The advantages of near-field ultrasonic phased-array imaging technology are as follows: the detection area and position can be defined flexibly; and the fixed-point measurement can be realized, which is especially suitable for detecting the damage position. Also, due to the high detection accuracy, when a few detection errors occur, the entire detection image reconstruction will not be affected. The specific flowchart demonstrating near-field damage imaging monitoring is shown in Figure 3.

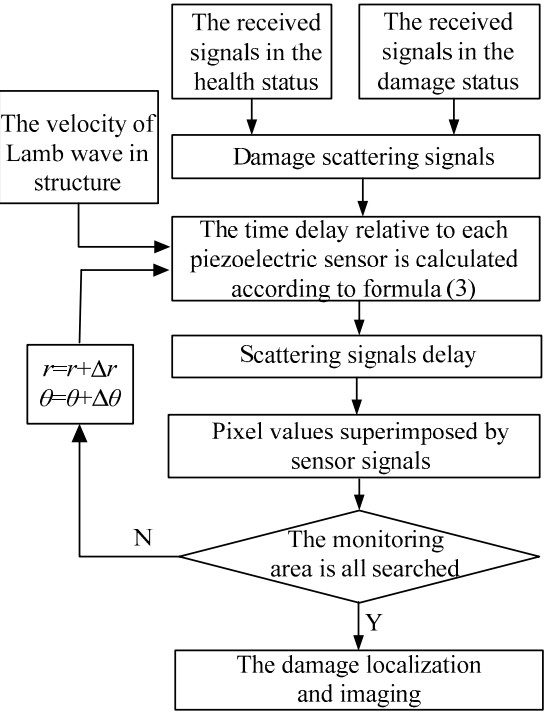

**Figure 3.** Flowchart of the near-field sampling ultrasonic phased-array damage imaging monitoring process.

The main shortcoming of near-field phased-array imaging is that in this method each polar angle $\theta$ (the angle step is 1°) is used in data collection, which makes data collection laborious. Moreover, the problem of angle blindness and false images can also occur in one-dimensional phased-array ultrasonic monitoring. Thus, the key current problem is how to realize omnidirectional near-field damage monitoring quickly and efficiently.

Through the experimental research, it has been found that the angle scanning realized by the time delay of the excitation signals of each array element in advance is as essential as the angle scanning realized by the time delay of the receiving signal of sensor elements in a later stage, which denotes the control of the initial phase of signals [12].

Therefore, in this work, we study the sampling phased-array data acquisition technology to improve data collection, where the viewpoints are not distinguished, and every piezoelectric element is used in the scanning process such that when one of the piezoelectric elements is in the incentive operating mode, the rest of the piezoelectric elements act as sensors. Then, using the already explained scanning principle and determination of the time delay of each of the elements, the overall sensing signal delay is determined. Sampling phased-array data collection is basically the same as traditional phased-array data collection but less time consuming and more efficient. In addition, the two-dimensional cruciform array can be used to realize the omnidirectional detection and solve the problem of angle blind areas and pseudo-images.

## 3. Experiments

### 3.1. Experimental Setup

In the experiments, we analyzed the composite materials for an aviation fuel tank design. The tank structure used is presented in Figure 4, where it can be seen that it contained upper and lower fuel tank parts that were both of the T300/QY8911 type with variable thickness in the carbon fiber composite panel on the fuel tank surface. The fuel tank had a carbon fibre composite structure on both ends of the middle thick thin. The thickness of the middle area was 7 mm, the thickness of both its ends was 4.5 mm, and the thickness of the middle area had an obvious transition. On all four sides, there were 7050-T7451 metal aluminum alloy plates, which were connected and fixed to the upper and lower surfaces by two-rows of rivets. Meanwhile, two ribbed structures were arranged on the left and right sides of the plates to enhance the structural strength. The overall size of the fuel tank was 600 mm × 300 mm × 240 mm (length × width × height).

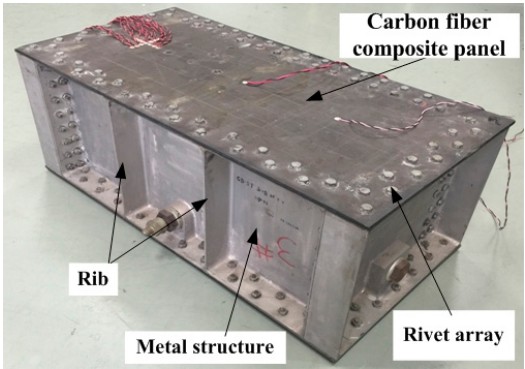

**Figure 4.** The fuel tank structure.

The experimental system is shown in Figure 5, where it can be seen that the experimental system contained the integrated piezoelectric multi-channel scanning system developed by Qiu and Yuan [15] mounted on the tank under the board surface, which consisted of a uniform two-dimensional cross array and had linear arrays I and II, whose centers were in the coordinate origin and were along the *x*-axis and *y*-axis, respectively. In the two linear arrays, the diameter and thickness of each of the PZT elements were 8 mm and 0.48 mm, respectively. Each of the linear sub-arrays of the cross array was made up of seven PZT elements with 9 mm spacing between two adjacent elements. The piezoelectric elements in linear array I were labeled from left to right as I-0, I-1, . . . , I-6, and the piezoelectric elements in linear array II were labeled from bottom to top as II-0, II-1, . . . , II-6. The discussion about the correct positioning and number of sensors for obtaining the efficient signals can be found in [9].

In the structural damage detection process, the integrated piezoelectric multi-channel scanning system was in the active mode, and the excitation signal was a five-wave peak narrowband sinusoidal modulation signal with an amplitude of ±7 V, and it was applied to the excitation piezoelectric element after being amplified by 10 times by a power amplifier. The central signal frequency was 50 kHz.

The sampling frequency was 4 MHz, including the pre-collection length of 2000 points, and the data sampling point length was 4000 points.

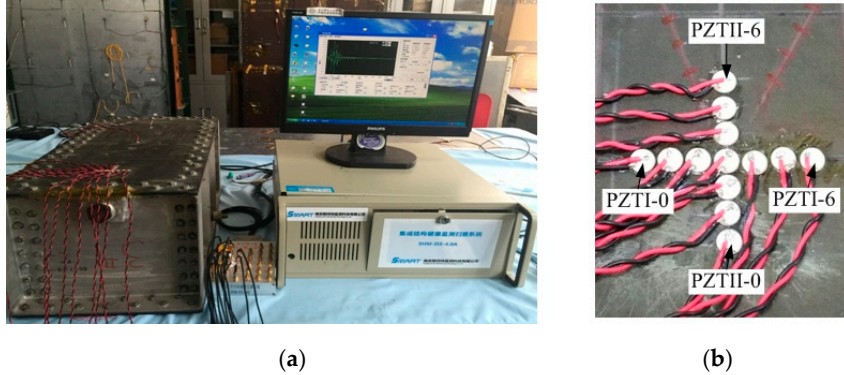

| (**a**) | (**b**) |

**Figure 5.** The experimental setup. (**a**) The experimental system (**b**) The cross-array and its numbered components.

### 3.2. Calculation of Aviation Fuel Tank Wave Velocity

Carbon fiber composite material is an anisotropic material. Therefore, the propagation speed of Lamb waves in different directions in a carbon fiber composite material is different. With the aim to identify the actual damage position in a composite material as accurately as possible, the propagation speed of Lamb waves in the oil tank was calculated first.

As it is presented in Figure 6, the cross-array center position was in the right-angle coordinate system origin, and arrays I and II were along the *x*-axis and *y*-axis direction, respectively; there were twelve piezoelectric sensors which were arranged on the left and right side of the array, and they were numbered from 1 to 12. A cruciform, central-positioned piezoelectric element was used for excitation, and the remaining twelve piezoelectric elements acted as sensors. In Figure 6, the dotted-line arrows denote the excitation-sensing paths during the wave velocity measurement, and they were used to determine the velocity of Lamb waves in the plate at a certain angle.

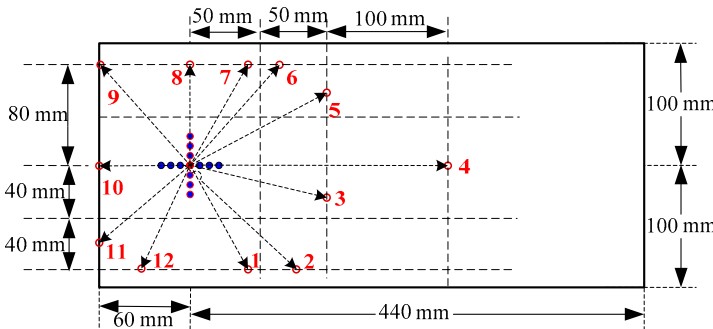

**Figure 6.** Schematic diagram of the piezoelectric sensor and energizing sensing channel.

The propagation velocity of Lamb waves in the plate was obtained by using the active excitation-sensing method. The twelve piezoelectric sensing elements and the central piezoelectric element in the array corresponded to 12 excitation-sensing paths, and the propagation velocity in all the corresponding 12 different directions was calculated. Using the Shannon continuous complex wavelet transformation method, the start time of the excitation signal and the arrival time of each sensor-response signal were measured, and the difference between these two times denoted the propagation time of the Lamb wave. Since the distance of the piezoelectric sensors from the coordinate system origin was known, the propagation velocity of a Lamb wave in the tank could be calculated for each piezoelectric element in each direction using the corresponding wave propagation time.

For instance, consider the piezoelectric sensor 4 (Figure 7) and the Lamb wave propagating in the 90°-direction. Using the first 2000 sample points and applying the Shannon continuous complex wavelet transformation method, we could calculate the starting time of the excitation signal and the arrival time of the sensing signal, respectively expressed as $t_0$ and $t_4$. The time difference between $t_0$ and $t_4$ denotes the time required by the Lamb wave to travel from the origin to the piezoelectric element 4, and it is given by:

$$\Delta t_4 = t_0 - t_4 = (1.835 - 0.5375) \times 10^{-4} = 0.1298 \ (ms) \tag{10}$$

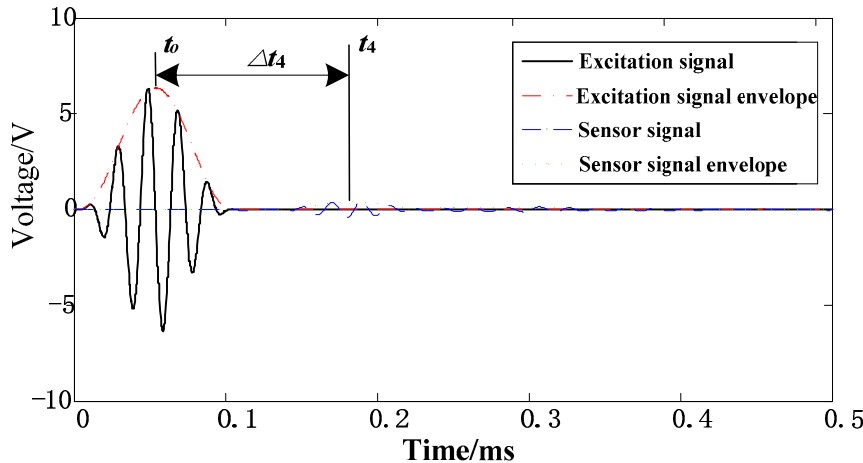

**Figure 7.** The propagation of the signal in the 90°-direction.

According to the calculated propagation time, the propagation distance $L_4$ was 200 mm. Therefore, the propagation velocity in the 90°-direction of a Lamb wave in the fuel tank of the carbon fiber composite material at the excitation of 50 KHz was calculated by:

$$c(90°) = \frac{L_4}{\Delta t_4} = \frac{200}{0.1298} = 1541 \ (m/s) \tag{11}$$

Similarly, the propagation velocity of the Lamb waves in the other eleven directions was calculated. The propagation time in all directions was obtained by subtracting the arrival time of the excitation signal from the corresponding arrival time of the peak value of the Lamb wave, the A0 wave packet. Finally, the propagation speed of the Lamb waves in all directions was obtained using the wave propagation distance. The specific propagation velocity values are given in Table 1.

**Table 1.** The velocity of Lamb waves at different angles for different elements.

| Element Number | 1 | 2 | 3 | 5 | 6 | 7 | 8 | 9 | 10 | 11 | 12 |
|---|---|---|---|---|---|---|---|---|---|---|---|
| Angle (°) | 32 | 40 | 80 | 120 | 140 | 160 | 180 | 206 | 270 | 316 | 345 |
| Arrival time (ms) | 0.11775 | 0.118 | 0.1185 | 0.1335 | 0.1185 | 0.10925 | 0.11175 | 0.11925 | 0.09574 | 0.10925 | 0.10925 |
| TDOA (ms) | 0.064 | 0.06425 | 0.06475 | 0.07975 | 0.06475 | 0.0555 | 0.058 | 0.0655 | 0.0555 | 0.0555 | 0.0555 |
| Distance (mm) | 91 | 110 | 102 | 106 | 106 | 85 | 80 | 99.4 | 60 | 84 | 85.4 |
| Velocity (m/s) | 1422 | 1712 | 1575 | 1329 | 1637 | 1532 | 1379 | 1518 | 1491 | 1514 | 1539 |

Using the velocity values presented above, the velocity of the Lamb wave in the tank of a carbon fiber composite material at the excitation of 50 kHz in twelve directions was fitted to the propagation speed curve shown in Figure 8. Given the anisotropy and variable thickness of fuel tank structure, the reflection signal is more complex. In addition, there will be errors in the measurement of the arrival

time of the sampled signal. The error is present according to the recent literature [16], and the damage location result is not affected due to the small measurement error of wave velocity anyway.

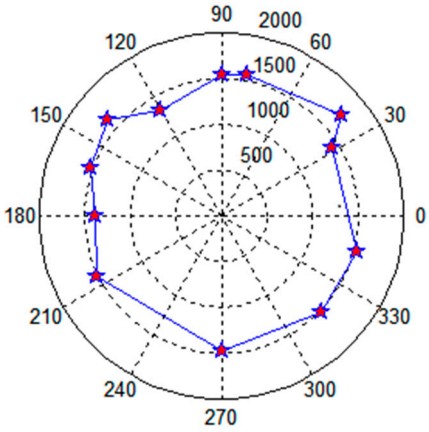

**Figure 8.** The tank speed curve.

## 4. Experimental Results Analysis

The positions of the piezoelectric elements and single damage point P are shown in Figure 9. Using 401 glue paste, a hollow hexagon screw with a diameter of 13 mm was used to simulate structural damage whose polar coordinates were $P$ (90 mm, 109°) and whose Cartesian coordinates were $P$ (−29.3 mm, 85.1 mm). Thus, according to (2), $P$ was located in the near-field region.

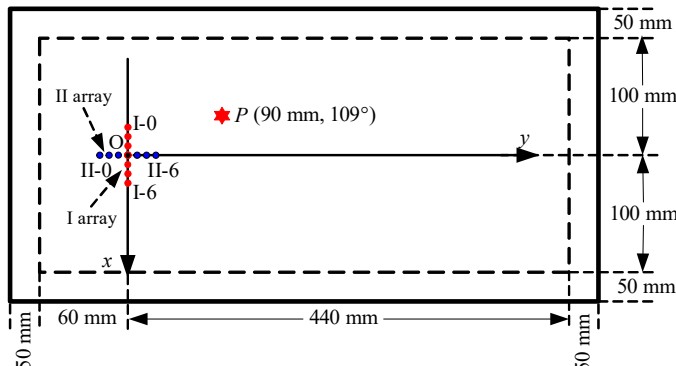

**Figure 9.** The locations of the piezoelectric elements and single damages.

Taking array I with the excitation at the piezoelectric element 0 and piezoelectric sensor 6 as an example, the influence of a tank structure on the Lamb wave signal propagation was analyzed. In the experiment, an excitation signal with a central frequency of 50 kHz was used, and the sampling frequency was 4 MHz. Additionally, 2000 points before the data sampling point were used in the calculation. The excitation signal of the piezoelectric element 0 is shown in Figure 10a, and the reflection signal of the Lamb wave received by the piezoelectric plate 6 is shown in Figure 10b.

As shown in Figure 10b, the received signal contained a lot of overlapping reflected signals from the Lamb wave signal, which resulted from the reflection from the bottom boundary. At last, there were mixed, reflected Lamb wave signals, which resulted from the reflection from the left and right boundaries above the array edge.

In Figure 10b, it can be seen that as the propagation distance increased, the attenuation of the Lamb wave increased relatively fast. Therefore, in order to identify the damage and make it easier to determine, we used the damage scattering signal; in other words, we used the difference between the sensor signal collected in the health state and the sensor signal collected in the damage state to analyze the structural damage.

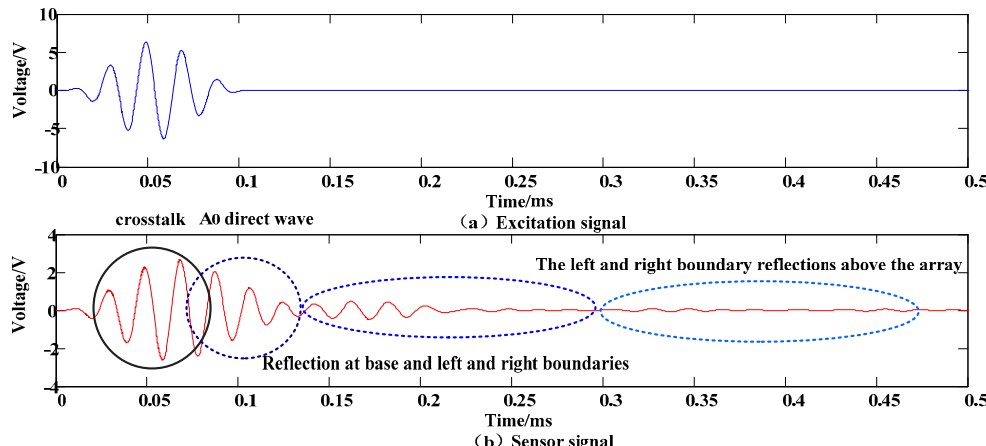

**Figure 10.** The excitation signal and the sensor signal.

As shown in Figure 11, in array I, the piezoelectric element 0 generated the excitation signal, and the other sixth piezoelectric elements received the signal. A received sensing signal containing no damage was considered as a health signal, as shown in Figure 11a. A received sensing signal containing any kind of damage was considered as a damage signal, as shown in Figure 11b. The difference between the health and damage signals denoted the damage scattering signal, as shown in Figure 11c. In Figure 11c, the front signal denotes the crosstalk signal, having the same position as the excitation signal. The damage scattering signal originated from the Lamb wave signal generated by the excitation signal in the plate and reflected by the damage. The final damage scattering signal in Figure 11c was produced by the Lamb wave signal generated by the excitation signal in the plate, then reflected by the rivet boundary, and lastly reflected by the damage.

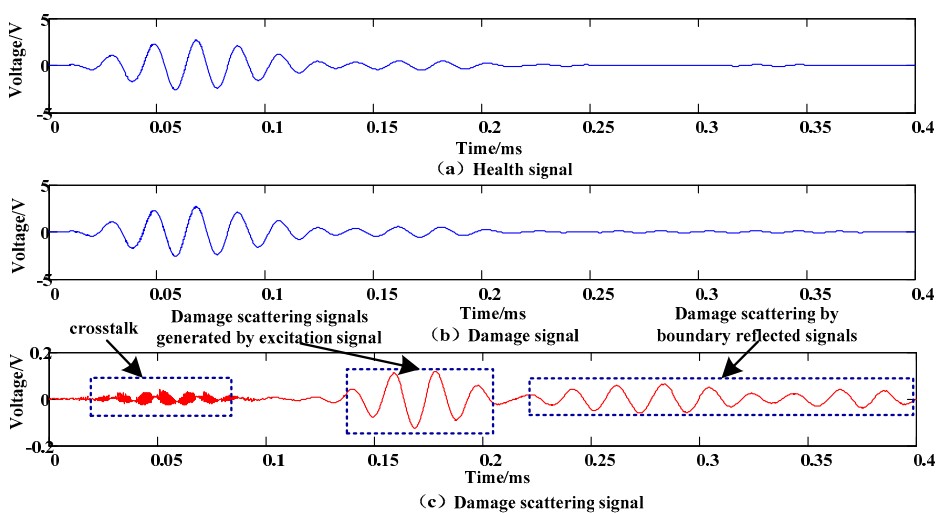

**Figure 11.** The sensor and damage scattering signals.

Data acquisition was only required to make all the array elements generate the excitation signal while the remaining elements were doing the sensing, without distinguishing the angle scanning. As mentioned previously, array I was along the *x*-axis direction, and array II was along the *y*-axis direction. During data acquisition, the health and damage signals were collected for each excitation-sensing combination made of seven piezoelectric elements, where one of the seven piezoelectric elements generated the excitation signal and the other six piezoelectric elements sensed and received the scattered signals generated in the structure.

The damage scattering signal of each piezoelectric sensor was obtained by subtracting the corresponding health signal and damage signal. The near-field phased-array sampling damage

imaging monitoring method presented in Figure 3 was used to determine the time delay of the damage scattering signals. Figure 12a shows the angle-time-amplitude imaging of the cross-array composite damage scattering signal; Figure 12b shows the more intuitive and easy to observe Cartesian coordinate imaging of damage. In Figure 12b, *x* and *y* are the horizontal and vertical axes in the Cartesian coordinate system.

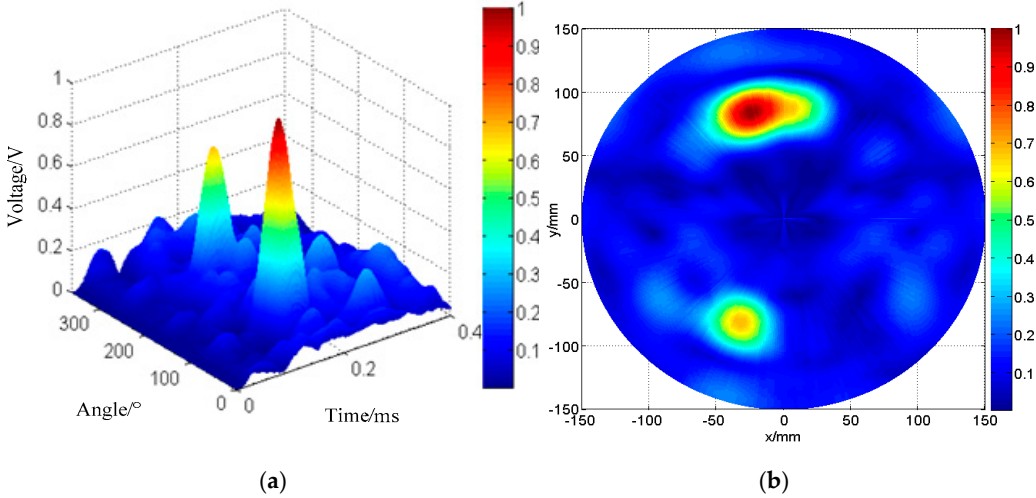

(**a**)　　　　　　　　　　　　　　　　　　　　　　(**b**)

**Figure 12.** Damage imaging. (**a**) Angle-time-amplitude damage imaging (**b**) Cartesian damage imaging.

The exponential function-based algorithm was employed to improve the scanned image of the material damage. As shown in Figure 13: the image enhancement improved the positioning accuracy of damage. The final measurement of damage location provided the Cartesian coordinates of *P* (−24.14 mm, 84.2 mm) and polar coordinates of (87.6 mm, 106°); while the actual Cartesian and polar coordinates of the damage were (−29.3 mm, 85.1 mm) and *P* (90 mm, 109°);, respectively, so the location error of 5.2 mm and angle error of 3° were achieved.

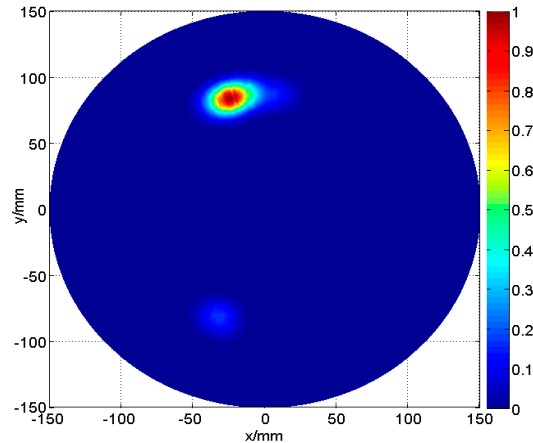

**Figure 13.** Damage imaging after the enhancement of the cross-array image.

## 5. Conclusions

This paper proposed an omnidirectional near-field sampling ultrasonic phased-array damage imaging method. The proposed method was verified by experiments using variable-thickness carbon fiber composite materials for the panels of fuel tanks. The experimental results showed that the proposed omnidirectional near-field sampling ultrasonic phased-array damage imaging method can achieve accurate positioning of material damage, providing the damage location error is about 5 mm, and has an angle recognition error of 3°. The proposed method can effectively solve the problem of

near-field blind areas and angle blind areas of traditional damage positioning methods, and realize more accurate damage localization.

**Author Contributions:** Z.W. developed the method. Z.W. and. Z.X. conceived of and designed the experiments. Y.L. and Y.J. performed the experiments. Z.W. wrote the paper.

**Funding:** This work is supported by the National Natural Science Foundation of China (No. 51605223), the Qing Lan Project of Jiangsu Province of China, and the High-level talent work start-up fee funded project of the Jinling Institute of Technology of China (No.jit-b-201823).

**Acknowledgments:** The authors would like to thank the Natural Science Foundation of China, the Qing Lan Project of Jiangsu Province of China, and the High-level talent work start-up fee funded project of the Jinling Institute of Technology of China for supporting this research and helping its realization.

**Conflicts of Interest:** The authors declare no conflicts of interest.

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
