# Peer review of "An Omnidirectional Near-Field Comprehensive Damage Detection Method for Composite Structures"

_applsci, doi:10.3390/app9030567_

Reviewer 1 Report

Did not see a discussion on the crucial issue of correct placement of sensors and number of sensors for efficient signals.

The Authors failed to mention with computer program or interface used to plot or interpolate the signals for 3D mapping.

Line 194:The Authors failed to identify the manufacturer of the piezoelectric multi-channel scanning system.

Author Response

Dear Reviewer:

Thank you very much for your help regarding our paper entitled “An omnidirectional near-field comprehensive damage detection method for composite structures”. The authors truly appreciate the time the referees spent for this paper. The authors have fully addressed these comments in the revised paper. The following upload file provides a detail point-to-point reply to main comments, and the relevant actions taken in order to improve the quality of this paper.

Reviewer 2 Report

The authors present an interesting approach for near field damage detection in composite structures through phased array technique. The paper is well written and organized and presents experimental results that are quite convincing.

However, before I can recommend the manuscript for publication in, I suggest the authors to address a few major comments listed below.

1. The phased array technique is proposed for monitoring approach. However the authors do not explain why they propose this approach over classic global reconstruction approaches available in the literature and able to detect and localize damages in composite structures using pitch catch technique [1, 2]. 

2. The technique is based on velocity assessment on several directions to then exploit the phased array technique based on time delay between several sensors. However when dealing with discrete signals, the arrival time cannot be assessed without a defined error, as recently demonstrated in [3]. The result in Figure 8 clearly demonstrates the presence of such an error in time of flight calculation. Along 90°/270* direction, where the velocity should be the same, the difference is indeed about 400 m/s. The authors are strongly encouraged to explain how they dealt with the time of arrival error. If they considered it, they should emphasize this aspect. Otherwise, they should mention at least that the error is present according to the recent literature and that the result is not affected anyway,

 As a minor comment, I strongly encourage the authors to proofread the article due to minor mistakes and simplify some sentences which reduce the readability of the paper.

 REFERENCES:

[1] Moll et al., Multi-site damage localization in anisotropic plate-like structures using an active guided wave structural health monitoring system, Smart Materials and Structures 19 (4), 045022, 2010.

[2] Memmolo et al., Multi-site damage localization in anisotropic plate-like structures using an active guided wave structural health monitoring system, Aerospace 5(4), 111, 2018.

[3] Maio et al., Application of laser Doppler vibrometry for ultrasonic velocity assessment in a composite panel with defect, Composite Structures 184, 1030-1039, 2018.

Author Response

Dear Reviewer:

Thank you very much for your help regarding our paper entitled “An omnidirectional near-field comprehensive damage detection method for composite structures”. The authors truly appreciate the time the referees spent for this paper. The authors have fully addressed these comments in the revised paper. The following upload file provides a detail point-to-point reply to main comments, and the relevant actions taken in order to improve the quality of this paper.

Round  2

Reviewer 2 Report

The authors did not address the comments reported in the previous review round.

They change the velocity along the direction 270° due to a previous error but they limited the comments in the response file. They are strongly suggested to detail such comments in the manuscript and address the issue about the velocity assessment before I can recommend the article for publication.

Author Response

Thank you very much for your help regarding our paper entitled “An omnidirectional near-field comprehensive damage detection method for composite structures”. The authors truly appreciate the time the referees spent for this paper. The following provides a detail point-to-point reply to main comments, and the relevant actions taken in order to improve the quality of this paper.

Round  3

Reviewer 2 Report

The authors detailed the issue about the wave velocity assessment evidencing that the method proposed is robust respect to such errors in velocity estimation.  

I recommend the publication of the paper in the present form.